# Two-Process Random Textures: Measurement, Characterization, Modeling and Tribological Impact: A Review

**DOI:** 10.3390/ma15010268

**Published:** 2021-12-30

**Authors:** Pawel Pawlus, Rafal Reizer, Wieslaw Żelasko

**Affiliations:** 1Faculty of Mechanical Engineering and Aeronautics, Rzeszow University of Technology, Powstancow Warszawy 8 Street, 35-959 Rzeszow, Poland; 2Institute of Materials Engineering, College of Natural Science, University of Rzeszow, Pigonia Street 1, 35-310 Rzeszow, Poland; rreizer@ur.edu.pl; 3Faculty of Mechanics and Technology, Rzeszow University of Technology, Kwiatkowskiego Street 4, 37-450 Stalowa Wola, Poland; w.zelasko@prz.edu.pl

**Keywords:** two-process surface, measurement, modeling, characterization, contact mechanics, friction, and wear

## Abstract

Two-process random textures seem to present better functional properties than one-process surfaces. There are many random two-process textures. Plateau-honed cylinder surfaces are the most popular example. Two-process surfaces are also created during the initial periods of life of machined elements. However, knowledge about two-process textures measurement, modeling, and behavior is low. Two-process surfaces are very sensitive to measurement errors. It is very difficult to model them. Special methods of their characterization were created. Their functional significance was studied in a small number of publications. In this paper, measurement, characterization, and modeling of two-process textures were presented. The functional impact of them was analyzed, the effects on contact mechanics and friction and wear were mainly studied. Finally, considerations of future challenges were addressed. The nature of two-process random textures should be taken into account during analyses of properties of machined elements. The plateau part decides about the asperity contact, and the valley portion governs the hydrodynamic lubrication.

## 1. Introduction

All surfaces are rough. Roughness affects various properties, such as contact, friction, lubrication, and wear [1]. Therefore, many researchers analyzed the functional properties of the rough surfaces. The majority of the research work was related to random surfaces of Gaussian ordinate distribution. For example, the statistical models of contact were related to Gaussian surfaces [2,3,4,5,6,7]. 

Initial methods of analysis, such as filtering, using Gaussian filters [8,9,10,11,12,13] of textures, were related to surfaces of symmetrical ordinate distribution. However, two-process surfaces and generally multiprocess textures seem to present better functional properties than one-process surfaces. The plateau-honed cylinder surface is the most popular example of two-process random textures. Cylinder liners are commonly made of gray cast iron. Due to the recent trend toward lighter engines, the materials for the engine blocks changed from cast iron to aluminum alloys. To improve wear resistance, many coatings, such as thermally sprayed coatings or Nikasil plating, are applied to aluminum bores. The plateau-honed cylinder has traces of two processes: final honing and plateau honing. During the final honing deep valleys are created, while during plateau honing the plateau smooth structure is formed. This surface consists of two parts, the plateau and the valley. Two-process surfaces are created during a mild wear of one-process textures—the resemblance to surfaces created during running-in is the first reason for plateau honing process creation. Campbell [14] revealed in theoretical investigations that obtaining the same material ratio during volumetric wear of a two-process surfaces would be smaller than that of one-process surface characterized by the same roughness height. Figure 1 illustrates this. It presents one- and two-process profiles characterized by the same rms. height Pq, their material ratio curves, and selected parameters. The material ratio curve (Abbott-Firestone) represents the cumulative height distribution [15]. For a one-process unfiltered profile maximum peak height Pp and maximum valley depth Pv are similar, in contrast to the two-process profile, for which the Pv parameter is higher than the Pp parameter. To obtain 40% of the material ratio, the wear volume of the two-process profile would be nearly three times smaller than that of the one-process profile Aw. It is related to a smaller wear and quicker running-in of the two-process surface. The Pv parameter describes the area under the profile material ratio curve [16]. One can see that assuming the same volumetric wear for which the one-process surface will vanish, the plateau part of the two-process surface will still exist. One can see also that the oil capacity Ac of a two-process surface is twice that of a one-process Ac texture. It will result in better tribological properties, such as a smaller risk of the seizure of two-process surface in starved lubrication conditions.

There are other possible advantages of two-process textures over one-process surfaces—smaller wear of the smoother peak part and the possibility of creating hydrodynamic lift by deep valleys [17]. However, after comparison of tribological performances of one- and two-process textures, contradictory results were obtained by various researchers. To correctly assess the functional behavior of two-process surfaces, these should be precisely analyzed. However, their description is difficult. Furthermore, two-process random surfaces are sensitive to measurement errors. Surface modeling led to a reduction in the time and cost of experimental investigations. Most of the work was devoted to the simulation of random two-process surfaces [18,19,20,21,22]. Two-process surface modeling is more difficult [21].

Since two-process surfaces are sensitive to measurement errors, problems related to these errors will be described. This topic is important, since measurement errors can lead to false prediction of functional properties of machine elements. Next, parameters developed specially to two-process surface description will be presented. The main task in random two-process texture characterization is its partition into plateau and valley parts. It will be presented in this paper as well as other topics related to characterization of two-process surface. Methods of two-process texture generation, mainly based on superimposition procedure will be also presented. Finally, the effects of two-process surfaces on contact mechanics, as well as on friction and wear, will be shown. Summarizing, the structure of this paper is as follows: measurement and characterization of two-process surfaces (measurement errors, specific parameters, characterization), simulation and impact (contact mechanics, friction and wear).

## 2. Measurement and Characterization of Two-Process Random Surfaces

### 2.1. Measurement Errors

Two-process surfaces are very sensitive to measurement errors. These errors can be divided into the following groups [23,24]:errors typical to the method of measurement,errors related to quantization and digitization,errors related to preprocessing such as filtration,other errors.

Data processing is completed after measurements have been taken.

The stylus method is commonly used in the measurement of areal (3D) surface topography. As a result of the long measuring time, it was typically replaced by optical methods. Among them, white light interferometer and confocal methods are the most popular.

When using a stylus, the main error is caused by the mechanical filtration of the tip of the stylus. The lateral resolution depends critically on the size of the stylus tip and, to a lesser degree, on the flank angle. The problem is that the radius of the stylus may not penetrate fully to the bottom of the valley. The tip of the stylus acts as a low-pass filter by cutting out high frequencies. The stylus can distort information of a wavelength 10 times larger than the diameter of the tip [23]. Figure 2 presents the mechanical filtration of the stylus tip.

The path of the spherical tip of the stylus was modeled, according to [25]. In this case, distortion of the profiles was analyzed on only one cross-section. The two-process profile consists of two parts: plateau and valley. Each part is characterized by amplitude parameters and the horizontal parameter. When the horizontal parameter of the valley portion was smaller than that of the plateau part, the changes in parameters due to mechanical filtration were larger than those obtained for the opposite case. This could be explained by the fact that the stylus tip could not go inside thin valleys. The changes in the standard deviation of the height (Pq parameter) of the two-process profile were substantially larger than those of the one-process profile, of similar roughness height. The changes in the profile slope were similar to the change in the Pq parameter. The horizontal parameters increased, but their changes were smaller than those of the Pq parameter. For cylinder surfaces after plateau honing, a mechanical filtration by 3D ball caused larger changes of amplitude distribution than that of 2D circle. It was due to the real width of the valleys—when the valley is inclined to the measurement direction, the real width of the valley is smaller than the width of the valley in the profile—so it is more difficult for the stylus tip to penetrate inside the valley [26].

The slower the stylus moves, the finer the details can be studied. At comparatively high speed, the stylus flight is possible—the stylus can lose contact with the surface due to the presence of a rapid impulse, such as a growing step. The speed of the stylus, the force of the stylus, the damping constant, and the surface characteristics affect the flight of the stylus. The effect of stylus flight is to record negative slopes as milder [27,28].

The experiment was performed using a Talyscan 150 measuring instrument with the following sliding speeds: 0.5, 1, 2, and 3 mm/s. After the measurement with the highest measuring speed, the height parameters decreased, but their changes were comparatively small; the maximum errors in the Sa, Sz, Sv, and Sq parameters were 10%. The changes in Sp could be greater (to 23%). The tendency of the emptiness coefficient to change depends on the character of the surface topography (for example, the width of valleys). When Sp/Sz decreased, the highest changes (decrease) of Sp occurred. When the Sp/Sz coefficient increased, the changes in Sv were larger than those in Sp. However, the biggest changes of the emptiness coefficient were 10% in comparison to the speed of 0.5 mm/s. The Sku parameter decreased during increasing speed, but its relative changes were not greater than 2%, The large reduction of rms. surface slope Sdq occurred (up to 20%). Summit density (up to 35%) and summit curvature decreased (to 38%). The hybrid parameter Sdr decreased, and its changes were the highest (up to 45%) [26,29]. Similar changes in profile parameters occurred. PSm and correlation length increased. Figure 3 shows the effect of the measurement speed on the parameters Sq and Sdq. One can see that for a speed of 2 mm/s these parameters increased compared to the slowest speed of 0.5 mm/s.

In the application of the white light interferometers and confocal measuring systems, when the light intensity obtained with a photodetector is too low, which is caused by high slopes, surface absorbance, or reflectivity, the surface points cannot be detected [30,31,32,33,34]. The non-measured points are typically replaced by a smooth shape computed from the neighbors, using various algorithms. The plateau-honed cylinder surface is very sensitive to errors caused by the presence of non-measured points, especially in the bottoms of valleys (see Figure 4)—even when the number of non-measured points is small, the errors of parameter calculation can be large.

Sharp edges cause the presence of outliers, called spikes, which are high and narrow peaks that do not really exist on the surface [35,36]. The spikes should be detected, and the points should be interpolated. Wang et al. [37] used several statistical methods for outlier detection.

The presence of spikes changes the shape of the material ratio curve; its peak part (left) has a vertical shape. Therefore, the presence of spikes caused mainly changes in the parameters characterizing the peak details of textures. The parameters characterizing the valley part, such as Sv, were stable. Other height parameters typically increased. The change in the Sa parameter was small. Changes in the Smr parameter were very large. The peak density Spd decreased, and the peak curvature Spc increased. Skewness Ssk and kurtosis Sku parameters typically increased. The changes in other parameters were rather low. Surface directionality Std was constant. Hybrid parameters, such as amplitude parameters increased.

Figure 5 shows a contour plot of the plateau-honed surface with and without spikes.

The process of analog-to-digital conversion, called digitization, depends on the representation of the analog signal by discrete data points. In the frequency domain, the ordinate values of the signal are recorded at equal sampling intervals. Quantization depends on splitting the signal into parallel height levels.

When sampling interval is too low, the data points are highly intercorrelated. On the other hand, the loss of spatial information can occur. In 3D surface topography measurement by stylus measurement, the sampling interval should be as large as possible to decrease the measurement time. The minimum sampling interval is similar to the dimension of the probe.

Several methods [38,39] were proposed to select the optimal sampling interval based on the acceptable tolerance between the parameters obtained from a sampled profile and their real values. The changes of hybrid parameters, summit density, and summit curvature are high with sampling interval change. Therefore, the sampling intervals were based on the surface spectral analysis [40,41,42]. Selection of the sampling interval depends on information on which wavelengths are the most important in surface functions. For example, according to Whitehouse and Archard [7] the sampling interval should be equal to the correlation length (the distance, at which the autocorrelation function decays to a given value, here 0.1), since after wear only long wavelengths exist on the surface. The problem of sampling interval selection is important for two-process texture. For a rougher surface, a higher sampling interval is required, while for the smoother surface, a smaller sampling interval is needed [43]. The sizes and density of deep valleys are functionally important. Therefore, Pawlus and Chetwynd proposed a sampling interval for which deep valleys can be identified [44]. Figure 6 presents the effect of the sampling interval equal to half of the width of the deep valley W, on the valley depth. For the best case, the obtained depth of the valley D should be H (real depth of the valley) and for the worst case it should be H/2. The sampling interval equal to 0.3 of the mean width of the deep valley was proposed.

To use this method, deep valleys should be identified. Various procedures can be used for this aim, typically on the basis of the material ratio curve, such as presented in [45]. The other method is based on the shape of the cumulative spectrum [42,44]. This approach can be used properly when only the valley part can be analyzed because the plateau part commonly contains a high-frequency component.

The quantization process depends on splitting the signal into a number of levels parallel to the surface. According to Whitehouse [46], the minimum number of quantization levels is 100. The authors of the articles [47,48] analyzed the influence of quantization on the characteristics of the 3D topography. The average roughness parameters varied steadily, but the skewness was very sensitive to changes in quantization levels. Quantization errors affect mainly parameters related to the peak surface part. The effect of the quantization error on the parameters of the two-process surface can be larger than that on the parameters of the Gaussian surface. For similar maximum heights, the quantization errors of the peak and valley parts of the Gaussian surface are the same, but for the two-process surface, the quantization errors affect mainly the peak surface portion—see Figure 7. Quantization errors can be monitored by the shape of the material ratio curve. Its stepwise shape proved that quantization errors occurred.

The peak density of 3D surface topography typically decreased, and the peak curvature increased due to quantization errors. The main directionality was constant. The correlation length and amplitude parameters were stable. Quantization errors can distort information about surface anisotropy.

The characteristic feature of two-process textures is that they contain deep valleys. These valleys are associated with problems of filtration. Previously, instruments used a filter that had a transmission characteristic similar to that of two capacitor-resistor networks connected in cascade, called a 2CR filter [8]. However, this filter led to phase distortion. Therefore, a Gaussian filter was established. Implementations of this filter were presented in [9,10,11,13]. Then, a Gaussian regression filter, working without running-in and running-out lengths was established. The Gaussian regression filter of 3D surface topography was developed [49,50]. Spline filters have properties similar to the Gaussian filter [51,52,53]. They work without marginal lengths. However, Gaussian and spline filters also can lead to distortion of the deep valleys. Phase distortion can be diminished by increasing the cut-off [54,55]. The alternative is to use no filter. Especially for plateau-honed surface textures, the valley suppression filter was established, earlier in the German standard DIN 4776 and then in ISO 13565-1 [56] standard. The filtering technique occurs in the following steps: the surface if filtered with a Gaussian digital filter to determine the mean line. Then all valleys below the mean line are removed. The profile is filtered again. The mean line obtained by the second filter (waviness) is superimposed on the original unfiltered profile. The filtering procedure leads to a more negative skewness and a higher kurtosis than that obtained after using the Gaussian filter [57]. Robust Gaussian filters were also developed [58]. They can be used for both one-process and two-process textures. For a one-process surface, their behaviors would be the same as those of Gaussian filters. The robust filter uses additional weights, which decrease in the place of peaks and valleys presence. They can work without marginal lengths in 2D and 3D systems. However, for a surface with very deep and wide valleys, robust filters should be modified [59].

After using an improper Gaussian filter for two-process profiles, the parameters describing the shape of the ordinate distribution changed. However, the changes in kurtosis and skewness were smaller than the changes in the emptiness coefficient Rp/Rt and the normalized core roughness depth Rk/Rt. The absolute values of the Rsk and Rku parameters decreased, while the emptiness coefficient increased. The horizontal parameter RSm decreased. Figure 8 presents the unfiltered profile, filtered profiles (roughness) after using Gaussian and robust Gaussian filters with material ratio curves and selected parameters.

Some morphological filters are also recommended for two-process textures, especially dilation, closing, and closing + opening filters. During a dilation, the structuring elements are in contact with the peaks of the surface (upper envelope), while in erosion with the surface valley (lower envelope). Shunmugam and Radhakrishman [60] first presented a direct dilation algorithm using the circle. Closing depends on dilation followed by erosion, and opening depends on erosion following dilation. Typically, the circle/sphere or horizontal line/plane is used as structural elements. Morphological filters were described by Dietzsch et al. [61], Krystek [62], Srinivasan [63], and Scott [64]. The authors of the articles [65,66,67,68] confirmed the applicability of morphological filters for two-process surfaces. After proper selection of the size of structuring element, the roughness profile will not be distorted. For example, Pawlus et al. presented a procedure for estimating the radius of the circular disc [68].

When filtering was not used, only the shape should be removed. It is typically undertaken by applying the polynomial. For two-process surfaces, using the low degree of polynomial is preferred. When the level of polynomial is too high, the distortion of deep valleys is possible. Specifically, the highest degree of the polynomial is three for two-process surfaces. Podulka et al. [69] presented the procedure to estimate the degree of the polynomial. It depends on selecting the degree for which the core roughness depth Sk achieves the minimum value. The better possibility is to exclude the valleys during form removal. However, it is not always possible, especially for the surface of plateau-honed cylinder liners.

### 2.2. Specific Parameters

Two standards ISO 135653-2 [70] and ISO 13565-3 [71] were dedicated to two-process random profiles. Both methods are based on the material ratio curve. The first group of parameters, called Rk, is based on the profile partition into 3 portions: valley, core, and peak. This division is executed by sliding a 40% of material ratio wide window through the entire curve. There are the following parameters: the core roughness depth Rk, the reduced peak height Rpk, the reduced valley depth Rvk, and two material ratios: Mr1 and Mr2 (Figure 9a) [72,73,74,75]. There is also the so-called oil capacity A2 dependent on the parameters Mr2 and Rvk. The material ratio Mr1 is of minor importance. This standard is based on earlier works by Trautwein [76,77].

The second ISO 13565-3 standard appeared in the USA. It is based on the partition of the two-process profile into 2 parts: peak (plateau) and valleys—Figure 9b [54,78,79,80]. It is based on the probability plot of the material ratio curve. In this plot, for one-process random profile, one straight line, while for two-process random structure, two straight lines are visible. The Rpq parameter (rms. height of the plateau part) is the slope of the upper straight line corresponding to the plateau part, while the Rvq parameter (rms. height of the valley portion) is the slope of the straight line that approximates the valley portion. The Rmq material ratio characterizes the plateau-to-valley transition. The depth of the plateau Pd, not included in standard, but important for profile modeling, is also presented in Figure 9b. It is a vertical distance between the mean lines of the plateau and the valley parts.

The mentioned parameters were extended to areal surface topography—ISO 25178-2 standard [81].

They were compared in [82]. Shortly speaking, the Rk/Sk parameters can be used for various surfaces, not only random two-process surfaces—which is their disadvantage [83]. They should not be used for deterministic surfaces; in those cases, there are problems with determining the parameters Rpk/Spk and Rvk/Svk. Furthermore, sometimes the implementation of this method can lead to false results [82]—the main problem is the dependence of the Mr2/Sr2 on the slope of the material ratio curve, sometimes leading to an increase in oil capacity during wear. The usefulness of the Mr1/Sr1 parameter is doubtful. The advantage of this method is the ease of parameter calculation, so it can be used for the assessment of the plateau honing process [84]. The areal parameters characterizing height are slightly higher than similar parameters describing profile. Since this method appeared first, it is the most commonly used. It can be combined with the areal volume V parameters also defined in the standard ISO 25178-2: Vmp—peak material volume, Vmc—core material volume Vvc—core void volume, and Vvv—dale void volume. The disadvantage of the V parameters is that they are based on arbitrary transition points between the peak and the core and the core and the valley [85].

The Rq/Sq family has a more theoretical background than the Rk/Sk group. It is based on two parts of the surface: the plateau and the valley. The areal parameters are similar to the profile parameters. It can be used for two-process texture modeling. However, its advantage is the difficulty in calculating the parameters. First, there is a problem with applying an approximating function of the probability plot of the material ratio curve. The other function that is easier to implement was proposed in [86]. The other problem is that the original procedure described in the ISO 13565-2 standard led to an overestimation of the Rpq parameter. This performance is related to the transition between the plateau and valley parts. Near this transition, overlapping between two Gaussian distributions occurred, this part having curvature should be removed during parameter calculation. In the procedure from the standard, the angle between the asymptotes of the approximating function should be bisected three times. It was found that further bisection decreases errors in the calculation of the Rpq parameter [87]. Recognition and joining of the details of the plateau part, based on [88], can also lead to improved calculation of the Rpq parameter (Figure 10). First, the valley edges should be determined. Next the plateau details should be joined. Recently, Sakakibara et al. [89] presented an improved algorithm to calculate parameters from the Rq group.

The probability parameters Rpq/Spq, Rvq/Svq, and Rmq/Smq can also be obtained from the height distribution [90,91,92,93,94].

### 2.3. Characterization of Random Two-Process Textures

The main problem in random two-process texture characterization is the partition of the surface into plateau and valley portions. It is related to the determination of oil capacity. The plateau and valley parts also affect differently the functional properties of two-process texture; therefore, the partition should be performed correctly. It can be undertaken using Rk/Sk and Rq/Sq groups; however, as was already said, the Rk/Sk family can lead to some errors. The transition point can be obtained as a point of maximum/minimum curvature (depending on the orientation of the vertical scale) of the normalized material ratio curve [95,96]. The additional method depends on rotating the curve of the normalized material ratio at an angle of 45 degrees and searching for the point of the highest ordinate (Figure 11) [82].

The transition point is helpful for the proper determination of the sizes of the deep valleys [95,97]. The parts of the plateau and valley can also be separated on the basis of morphological filtering [98]. The identification of deep valleys was performed by scientists [99,100,101,102]. Other methods for characterization of the plateau-honed cylinder texture can be found in the review paper [103].

As was mentioned in the Introduction, two-process textures are characterized by a negative value of skewness, and the emptiness coefficient smaller than 0.5. For two-process textures, negative skewness corresponds to large kurtosis. In addition, the Sp/Sv (Rp/Rv) ratio is less than 1. Kurtosis is proportional to the Sq/Sa (Rq/Ra) ratio [104], which is equal to 1.25 for surfaces of Gaussian ordinate distribution. Yousfi et al. [105] described the plateauness of the Spq/Svq (Rpq/Rvq) ratio, which is smaller than 1 for two-process surfaces. Article [106] contains other parameters that differentiate between one-process and two-process textures.

Generally, the ordinate distribution and the material ratio curve contain the same information. For the unimodal height distribution, the height corresponding to the maximum of the ordinate distribution corresponds to the smallest slope of the material ratio curve. Two-process random textures are superimpositions of two Gaussian surfaces. Therefore, these surfaces are sometimes called bi-Gaussian surfaces [92,93,94]. From intuition, these surfaces should be characterized by a bimodal height distribution. However, the probability height distribution of two-process surfaces is rarely bimodal. In [107] the limiting conditions of the bimodal height probability distribution of the two-process surfaces were developed. When the Smq parameter is greater than 50%, a unimodal amplitude distribution occurs. The ordinate of mode and the smallest slope of the material ratio curve correspond to the material ratio of 50% (Figure 12a–c). However, this ordinate for the unimodal height distribution and the Smq parameter smaller than 50% corresponds to that of the Smq parameter (Figure 12d–f). For the bimodal height distribution, typically the upper peak (Figure 12g–i) can be the main mode.

In the production process, surface roughness is typically assessed using only one parameter, the average roughness height Ra is commonly used. However, the structure of two-process surfaces is complicated and should be described by a set of parameters. The selected parameters should be functionally important and have low sensitivity to measurement errors. The analysis of the correlation between the parameters is substantial. The parameters that describe the surface should be uncorrelated and should characterize various surface features. Nowicki [108] and Gorlenko [109] used the linear correlation coefficient to select parameters that describe surface profiles, while Qi et al. [110] characterized areal textures. Fecske et al. [111] studied the correlation between the height parameters of the modeled surfaces of the Gaussian ordinate distribution. However, typically correlation analysis was used for surfaces after the same kind of machining. Terry and Brown et al. [112] analyzed ground surfaces, Ham and Powers [113] surfaces after single-point experimental forming, Reizer et al. [114] textured surfaces and Etxeberria et al. [115] surfaces from biomaterials. Many works on this topic were related to two-process textures [45,106,116,117,118,119,120]. The following parameters were selected for the description of the cylinder liners of the plateau-honed [45] and worn cylinder liners [116]: amplitude parameter, distance between deep valleys, and two parameters that describe the shape of the ordinate distribution, such as Rp/Rt and Rk/Rt. Pawlus et al. [106] recently revealed that the parameters Rq/Ra and Rp/Rt characterize the shape of the ordinate distribution of two-process profiles. Rosen et al. [117] revealed that the parameters describing the plateau region are more interrelated than the parameters describing the valley part. The correlation between parameters within the Sk/Rk and Sq/Rq groups of slide-honed cylinders was studied by Pawlus et al. [118]. They found that the parameters Rpk and Rk were intercorrelated; they proposed three parameters from the Sq/Rq group and Sk/Rk, Svk/Rvk, and Sr2/Mr2 parameters from the Sk/Rk family. Pawlus et al. [119] found after analysis of the modeled profiles that the parameters Rpk, Rk and Rpq parameters, Rvk and Rvq parameters, as well as Mr1, Mr2 and Rmq parameters were intercorrelated. Grabon and Pawlus [120] added to the parameters Spq, Svq, Smq or to Sk, Svk and Sr2 the following set of parameters: Sz, Ssk, Sal, Str, and mean curvature of the summits [121,122] describing various random two-process textures.

## 3. Simulation of Two-Process Random Textures

Typically, random 2D profiles or 3D surface topographies of the Gaussian ordinate distribution are modeled. For profile modeling, the input data are rms. height Rq and correlation length CL, which is the distance at which the autocorrelation function decays to an assumed value, typically 0.1. The shape of the autocorrelation function should also be specified. However, areal isotropic or one-directional anisotropic surface textures are characterized by the Sq parameter and correlation lengths in perpendicular directions. More complicated surfaces, such as cross-hatched surfaces after a one-process honing, are characterized by additional parameters, such as the honing angle.

Modeling of Gaussian textures is based mainly on time series ARMA and FFT methods. Among time series methods, typically AR (autoregressive) methods were used for profile modeling [123,124,125,126,127] and areal surface topography modeling [128,129,130,131,132]. Wu [18] and Newland [133] developed a method based on Fast Fourier Transform. Other methods of Gaussian texture modeling are presented in [21]. Bakolas [134] presented a method of simulating an oriented surface. Methods of generation of surfaces of non-Gaussian ordinate distribution are based on the Johnson translation system [135,136,137,138,139]—the skewness and kurtosis are additional input parameters. However, this method allows us to correct the simulation of textures with a low absolute value of skewness.

The two-process surface texture modeling procedure is related to the description using parameters from the Rq/Sq group. For profile generation, it is necessary to model two profiles. Each of these profiles is characterized by the Rq parameter and correlation length. However, the Rq parameter of the upper (plateau) profile is equal to the Rpq parameter of the two-process profile, while the Rq parameter of the lower (valley) profile is equal to the Rvq parameter of the two-process profile. The vertical distance between these profiles Pd (Figure 9b) is related to the parameters of the Rq group by the following equation:Pd = Rmq (Rpq − Rvq)(1)

The simulation of a two-process surface profile depends on the superimposition of two Gaussian profiles. It is based on selection from two Gaussian profiles points of smaller ordinates [140]. The iterative procedure was used to obtain the desired correlation length of the two-process profile. It depends on changing the correlation lengths of the plateau and valley profiles and selecting those profiles for which the correlation length of the two-process profile is the closest to the assumed value [141,142]. Figure 13 presents an example of the generation of a two-process profile.

One can predict the value of the the Rq parameter of two-process profile on the basis of the Rpq, Rvq, and Rmq parameters:Rq = RpqRmq + Rvq (1 − Rmq)(2)

Rmq should be given on a linear scale. The relative error of the prediction of the Rq parameter was found to be equal to 5.5%. Higher errors correspond to larger Rvq/Rpq ratios. Figure 14 presents two-process profiles with the Rq parameter calculated and predicted using Equation (2) Rqpr.

This procedure is helpful for comparing functional behaviors of one-process and two-process profiles characterized by a similar value of the Rq parameter.

This procedure can be extended for the modeling of the random surface areal topography of two-process random surface—Figure 15.

The procedure of simulation of a two-process structure after plateau honing is more complicated. It is based on creating cross-hatched Gaussian structures of parts of the plateau and valleys with their superimposition [140]. Some errors of the honing treatment can also be modeled.

Two-process surfaces can also be modeled on the basis of skewness and kurtosis using the Johnson translation system [135,136,137,138,139] or other procedures, such as those presented in [21]. However, skewness and kurtosis are not robust parameters sensitive to measurement errors in contrast to parameters from the Sq/Rq family [26]. Briefly, the same parameters Spq, Svq, and Smq could result in different values of the skewness Ssk and the kurtosis Sku.

There are different variants of the superimposition procedure. The simulated surface should be superimposed on the real surface or vice versa, two real surfaces can also be superimposed. The first possibility is used to simulate surfaces after low wear (smaller than the initial roughness height) [114,143,144]. During modeling, the one-directional surface of normal ordinate distribution was superimposed on the surfaces after machining. This method gave better results than the previously used truncation by a plane/line [145,146,147,148,149].

It is possible to obtain information on surface parameters after machining based on the measurement of the two-process surface (after machining and wear) [150].

## 4. Impact of Two-Process Random Textures

Many research works in the tribology field are related to behaviors of one-process surfaces. However, there is a comparatively small number of works that are concerned with two-process random surfaces. These surfaces are not only machined, including the most popular plateau-honed surfaces, but two-process surfaces are also formed in low wear, particularly during the running-in process of machine elements. The functional behavior of the surface created during running-in is better than that of the machined texture. Seal surfaces [151] and surfaces created by additive processes [152] are other examples of two-process textures. Generally, the plateau part decides about asperity friction, while the valley part about hydrodynamic friction. In this section, the effects of two-process surfaces on contact mechanics, as well as friction and wear, will be presented.

### 4.1. Contact Mechanics

Some works in the field of contact mechanics are related to surfaces of negative skewness that are typically associated with an improvement of contact characteristics [153,154,155,156,157,158,159]. In [153] the contact between two textures was modeled as a contact between a rigid flat and an elastic–plastic rough texture considering the work hardening and the interaction between the summits. Belhadjamor found that a negative skewness led to an improvement of contact stiffness. In [154] a contact of steel-on-steel surfaces was studied. Skewed surfaces had higher tangential stiffness, compared to Gaussian surfaces. Negative skewness led to more elastic contact [155,156]. Change and Jeng [157] revealed that skewed surfaces tended to improve contact and lubrication of smooth textures. McCool [158] found that a negative skewness led to a lower mean effective pressure than the Gaussian surface with the same value of the Sq parameter. Yu and Polycarpou [159] found that a surface with negative skewness resulted with a smaller area of contact, number of contacting asperities, contact load, and mean asperity pressure than Gaussian surfaces. Chilamakuri and Bhushan [135] revealed that a surface with a negative skewness caused a high contact area. Tayebi and Polycarpou [160] analyzed a contact between a magnetic disk of 2.4 GPa hardness and a slider of 22.6 GPa hardness. They found that negative skewness led to a higher static friction coefficient compared to the Gaussian case. To obtain low adhesion, a surface with positive skewness is needed [161,162].

Peng and Bhushan [163] analyzed a contact between a rough surface of the Al_2_O_3_-TiC head and a flat diamond tool. They revealed that the contact area of two-process surfaces with negative skewness decreased with an increase in the Rpq/Rvq ratio. However, the depth of the plateau has a marginal effect on the contact area and maximum contact pressure.

Tomanik [164] calculated the elastic contact area, asperity contact pressure, and surface separation on the basis of the measured surfaces of honed and plateau-honed cylinder liners made of cast iron. He considered only summits to exist above the mean surface plane.

Leefe [165] analyzed the elastic contact of two-process random surfaces. He found that the plateau surface part governed the contact performance of the two-process texture. He also found that the cumulative distribution of the asperity height can be approximated by two straight lines, similar to the cumulative distribution of surface ordinates—Figure 16. Leefe studied many worn surfaces, for example, from worn seal rings, mainly carbon graphites and ceramics.

Hu et al. [166] studied the influence of the parameters Spq, Svq, and Smq on the parameters that characterize the summits, such as rms. height, density of the summits, and mean radius of the curvature of summits. Rms. height of the summits increases when Spq and Svq increase and Smq decreases. However, rms. height of the summits located in the upper part of the surface part depends only on the Spq parameter. The increase of a Smq parameter means the higher proportion of upper components, located above the knee-point—Figure 16, therefore the higher Smq had a positive effect on the summit density of the upper part. The radius of summit curvature is positively related to the Smq parameter, but negatively to the Spq parameter—see Table 1. Hu et al. [166] modeled worn surfaces, made of silicon carbide (SiC), tungsten carbide (WC), resin-impregnated carbon (RIC) and metal-impregnated carbon (MIC) from mechanical face seal.

σ_s_, σ_2ps_—standard deviation of summits height of entire surface, and the peak part, respectively, Sd_s_, Sd_2ps_—density of the summits of the entire surface and of the peak part, respectively, R, R_2p_—mean radius of summits curvature of entire surface, and of peak portion, respectively.

Pawlus et al. [167] analyzed a contact of steel-on-steel random two-process surfaces. In not all cases, the cumulative distribution of asperity heights can be approximated by two straight lines. When the correlation lengths CL are small (the ordinates of neighboring points are not correlated with each other), only one straight line is visible—Figure 17b. CLp and Clv mean the correlation length of the plateau and valley parts, respectively.

Greenwood and Wiliamson [168] developed the plasticity index to characterize the ability of the surface to plastic deformation.
(3)$$\Psi =\frac{E^{\prime }}{H}{\left(\frac{\sigma_{S}}{R}\right)}^{\frac{1}{2}}$$
where *H*—hardness of the softer material, *E′*—equivalent Young’s modulus.
(4)$$\frac{1}{E^{\prime }}=\frac{1-v_{1}^{2}}{E_{1}}+\frac{1-v_{2}^{2}}{E_{2}}$$
where $E_{1},\;E_{2},\;v_{1},\;v_{2}$ and Young’s moduli and Poisson’s ratios of two contacting surfaces, respectively. According to Greenwood and Williamson [168] the contact is elastic for the plasticity index smaller than 0.6 and plastic for the plasticity index higher than 1, while for the range 0.6–1 the mode of deformation is doubtful. Greenwood and Williamson analyzed the cumulative summit height distribution of specimen from worn mild steel.

Pawlus et al. [167] modified this version, considering only summits existing on the plateau surface part.
(5)Ψ2p=E′H(σ_2ps/R_2p )^(1/2)

The modified plasticity index, given by Equation (5) is much smaller than the plasticity index calculated for the entire surface—Equation (3). The errors in the determination of the plasticity index for the two-process surface are higher for a smaller Smq parameter, a longer Svq/Spq ratio and a higher correlation length of the valley surface. However, this index is larger than that obtained for the plateau Gaussian surface. It is caused by the fact that, when calculating the modified index, the highest point of the summits belongs to the peak part, but the other points can belong to the valley region. The plasticity index (4) is valid for isotropic surfaces. It was modified for the plateau-honed cross-hatched anisotropic texture made of gray cast iron [169], following the paper [170].

The authors of the papers [93,171,172] analyzed the contact of two-process random surfaces with a smooth flat using a deterministic approach. Contact of steel-on-steel flat surfaces was considered in [171,172]. Hu et al. [93], similar to [166] modeled worn surfaces from a face seal. The plateau part decides about the contact performance of two-process surfaces; however, the impact of the valley surface portion also exists [171]. For the same values of the Sq parameter of the one-process surface and Spq parameter of the two-process texture, the contact area is lower for one-process surface.

For plastically deformed surfaces, the effect of the sampling interval on the dependence between the contact load and the real contact area is negligible [172]. For surfaces inclined to plastic deformation, for the same contact pressure p, the contact area A increases with increasing sampling interval—Figure 18.

Zelasko [173] analyzed the contact of two-process random surfaces with a smooth sphere. The maximum contact pressures were smaller for two-process surfaces, when the Sq parameter of the one-process texture was equal to the Spq parameter of the two-process surface, the difference was higher for the larger Svq/Spq ratio of the two-process texture.

The majority of research in the field of contact mechanics is analytical. In [174] the plastic contact of steel surfaces with a hard flat surface was experimentally studied. The plastic deformation d was found to be proportional to the plasticity index calculated by Equations (3) and (4). The results were compared with those predicted using the elastic-plastic model JG developed by Jackson and Green [6]—Figure 19.

### 4.2. Friction and Wear

Similar to contact problems, several researchers obtained good tribological behaviors of negatively skewed surfaces. Sedlacek et al. [175,176] studied the contact between 100Cr6 steel discs of different roughness and ball made of Al_2_O_3_. They obtained the reduction in friction of negatively skewed surfaces in lubricating sliding.

Dzierwa et al. [177] analyzed sliding contact between disc made of 42CrMo4 steel of hardness 40 HRC and ball made of 100Cr6 steel of 62 HRC hardness [177]. He achieved a decrease in wear under dry contact of negatively skewed surfaces of small and high amplitudes, respectively. Jocsak et al. found after simulation that for the same surface height, the negative skewness of the cylinder liner surface reduced the friction of the piston ring pack [178]. Michail and Barber [179] found that skewed cylinder liners produced a smaller oil film thickness than the surface of normal ordinate distribution. However, this effect is minimized because of the typically lower roughness height of the skewed surface (for smaller amplitudes, the ratio of oil film thickness to roughness height increases).

The surfaces analyzed in References [175,176,177,178,179] were not highly skewed.

However, the textured surfaces are highly skewed. Kang et al. [180] after simulation found that under elasto-hydrodynamic (EHL) lubrication the change of the skewness from −0.5 to −2 for the same roughness height did not change the pressure and thickness of the film. However, decreasing the skewness to −4.7 caused significant increases in the oil film thickness and percentage of higher pressure.

Of course, random textures belong to a group of textured surfaces, but most of the textured surfaces had a random-deterministic character. Surface texturing is an option to improve the tribological properties of sliding surfaces by creating dimples (oil pockets and cavities). These dimples led to friction reduction mainly in mixed and boundary lubrication, a decrease in abrasive wear, and a decrease in the tendency to seizure. Laser texturing is the most popular technique for forming dimples. Reviews on surface texture are presented in References [181,182,183,184,185].

Jeng and Gao [186] found that the change in negative skewness was smaller during low wear than the change in positive skewness. Skewness typically decreased during wear. For high negative skewness, initially this parameter can increase.

Goeke et al. [187] achieved a smaller friction of the milled and honed surfaces in the lubricated reciprocating motion compared to those of the milled, milled and ground and polished surfaces. Surfaces after milling and grinding and after milling and honing were characterized by a similar roughness height, but the plateauness of the milled and honed surfaces was higher. Textures were created on the 18CrNiMo7-6 case hardening steel of 63 HRC hardness.

Pawlus found using the engine test bench that wear during running in was proportional to the emptiness coefficient Rp/Rt. He tested one- and two-process cylinder surfaces made of gray cast iron [188]. Smaller wear on the two-process surface than on the one-process textures was also received for wear higher than the initial surface height under artificially increased dustiness conditions [189]—Figure 20.

Jeng [190] compared the tribological behaviors of one- and two-process surfaces characterized by a similar value of the Rq parameter in lubricated conditions. The discs made of gray cast iron typically used as cylinder liner material co-acted with pin made of chrome-plated gray cast iron commonly used as a material of the first piston ring in automotive engines. Jeng found that the two-process surface was characterized by a higher resistance to scuffing. This behavior was probably caused by the higher oil capacity than one-process surface—see also Figure 1. Both surfaces behaved similarly under fluid lubrication. However, under mixed lubrication, the plateaued surface reduced the coefficient of friction, Figure 21a. This performance was caused by the fact that the valley surface part had a negligible effect on the asperity contact. Initially, the wear of the two-process surface was higher than that of the one-process surface, but the plateaued surface quicker finished running-in and its wear in the steady state period was lower—Figure 21b.

Barber and Ludema [17] conducted similar research. However, they did not achieve the tribological superiority of a two-process cylinder over one-process one.

Grabon et al. found in an experiment conducted using a test rig that the wear of the two-process cylinder was smaller than the wear of the one-process cylinder described by the same value of the Sq parameter [191]—Figure 22. Specimens were made from cylinder liner honed surfaces of grey cast iron of hardness 218 HB, the counter-specimens were made of chromium-coated steel C45.

The comparison between the tribological behaviors of one-process and two-process surfaces was the most correct when rms. heights of them were similar such as in References [189,190,191]. However, typically the amplitude of two-process texture is lower than that of one-process surface. For example, in [192] specimens were cut, specimens were cut from grey cast liners, and counter-specimens were made from a chromium-coated compression ring. A lower wear of the plateaued cylinder liners was achieved. On the contrary, Santochi and Vignale [193] obtained a higher amplitude of the two-process cylinder surface made of cast iron compared to the one-process surface. They achieved better operating parameters (power, fuel consumption) of plateaued cylinder texture. Similar, Yin et al. [194] obtained a smaller friction torque for the cylinder texture with a higher negative skewness and a higher amplitude. Sato et al. [195] found that a two-process cylinder liner made of cast iron improved the lubrication of Diesel engine in the beginning of running-in for comparatively high roughness (Rt near 2 µm). However, for low roughness height (Rt about 0.5 µm) one-honed liner led to more hydrodynamic lubrication. Yousfi et al. assessed the plateauness of the cylinder liner surface using the Spq/Svq ratio. Simulation revealed that a lower friction was related to a lower value of this ratio [196]. However, in experimental research the coefficient of friction was not correlated with the plateauness of the cylinder [105].

Changes in parameters of the Sq family during low cylinder wear are interesting. The Spq parameter of the initial two-process liner surface decreased, the Svq parameter was constant, while the Smq parameter increased. The one-process texture changed to two-process texture and the tendency of changes in parameters was similar to that mentioned above [192]—Figure 23. A similar tendency was found in other works in lubricated conditions [114,143]. Probably, after finishing low wear, two-process surface became one-process texture.

Discs of a 42CrMo4 steel of hardness of 40 HRC were put in contact with balls of 100Cr6 steel ball of a hardness of 60 HRC [197]. In lubricated sliding, the friction force of assemblies with two-process surfaces was smaller than that of one-process surfaces of the same amplitude. The coefficient of friction was proportional to the Sq parameter of one-process surface and the Spq parameter of two-process surface—Figure 24.

Until now, the comparison between the tribological performances of the one- and two-process surfaces has been presented under lubricated conditions. Only a few works on this topic have been carried out in the dry sliding regime. The results depend on the applied normal load. A 40 HRC hardened steel disc of hardness 40 HRC co-acted with steel pin of 64 HRC hardness [198]. When the normal load was 10 N and the test duration was 30 min, for smoother textures (Sq near 1 µm) wear volume of the two-process disc surface in the dry unidirectional siding was smaller than that of the one-process texture. The opposite situation occurred for rougher one- and two-process surfaces. In both cases, the distances to obtain the steady state of the coefficient of friction were larger for two-process textures. For the normal load of 20 N, the wear volume of the two-process surfaces was smaller than that of one-process surface also for the Sq parameter of 3 µm. However, for a normal load of 50 N and a test duration of 10 min, one- and two-process surfaces led to similar tribological performances [197].

In the works [197,198] one-process random surfaces of isotropic character were created by vapor blasting. Two-process random textures were formed by vapor blasting followed by lapping. They had also isotropic character.

Hu et al. analyzed changes in bi-Gaussian surface parameters in dry sliding [199,200]—Figure 25. For the hard material, the Spq parameter changed to the value obtained during running-in and the plateau part went down. When the low-wear process was finished, the plateau part went up due to the creation of deep scratches (the Svq parameter increased). For the soft material, initially the upper component went down and sharply went up because deep scratches were created. The Spq parameter changed to the value determined during running in. In a stable period, the Spq and Svq parameters were steady. This procedure was modified in [151]. The silicon-carbide sample was placed in contact with the carbon-graphite sample [151,199,200].

There are opinions that two-process surfaces have bi-fractal structures [201,202]. Wei et al. [203] developed bi-fractal characterization method used for the evaluation of a plateau-honed cylinder profile during the wear process.

Acoustic emission can be used for the analysis of the tribological interaction between sliding surfaces [166]. Huang et al. [204] developed a bi-Gaussian acoustic emission model for sliding friction of two-process textures. They analyzed the mechanism by which bi-Gaussian stratified topographies influence wettability [205,206,207].

Cylinder bores studied in [201] were made of gray cast iron. Similar to the works [166] and [93], worn surfaces made of silicon carbide (SiC), tungsten carbide (WC), resin-impregnated carbon (RiC) and metal-impregnated carbon (MiC) from a mechanical face seal were analyzed in [202,204]. The surfaces of steel discs made of AISI 52,100 were studied in [205,207]. The surfaces analyzed in [206] were fabricated by 3D direct-laser lithography attaching IP-S photoresist to an indium tin oxide-coated fused silica by means of a two-photon polymerization.

There are also three-, four- and generally multi-process surfaces, when the number of processes creating traces on textures is higher than two. Surfaces after plateau honing and wear made of gray cast iron can be an example of this texture [208]. The other possibility is presented in [209].

## 5. Conclusions and Perspectives

There are many random two-process textures. The plateau-honed cylinder surface is a typical example. Two-process surfaces are also created in low wear. Seal surfaces and surfaces formed in additive processes are other examples;The two-process surface consists of two parts: plateau and valley. These surfaces are more complicated than one-process textures. Therefore, the nature of two-process surfaces should be taken into consideration during studies of the properties of machined elements. The modeling of two-process surfaces is helpful during analysis;Two-process surfaces are sensitive to errors of measurement, especially the mechanical filtration of the stylus tip, the presence of non-measured points, the effect of sampling interval, and quantization error;Improper filtration causes distortion of two-process surfaces. The Gaussian filter should not be applied for these surfaces. Valley suppression filter, robust or morphological filters are preferred. Increasing a cut-off from 0.8 to 2.5 mm leads to the reduction of valleys distortions. Polynomials of low order are recommended to remove curvatures;Some surface topography parameters were specially designed to two-process random textures. Of these, the Rq family of parameters is preferred. The Rpq, Rvq, and Rmq parameters are statistically independent, based on theoretical background, and usable in surface modeling. However, the procedure for determining parameters is difficult to interpret numerically;One of the main problems in the analysis of two-process random surfaces is partition into plateau and valley parts, which differently affect functional properties. It is related to the determination of oil capacity. It can be undertaken using the various methods such as using Rq/Sq family, rotation of the material ratio curve, and searching for a point of minimum/maximum curvature;The ordinate distribution of random two-process surfaces is typically unimodal, especially when the Rmq/Smq parameter is greater than 50%. For the bimodal height distribution, usually the upper peak is the major mode;The analysis of linear correlation and regression is helpful in parameter selection. The basic description of surface topography of two-process surfaces should contain non-correlated parameters, describing various surface properties;The simulation of a two-process random surface depends on the superimposition of two Gaussian surfaces. One can predict the value of the Rq parameter of the two-process profile on the basis of the Rpq, Rvq, and Rmq parameters;Two-process surfaces are functionally important. Their tribological impacts are the most substantial under mixed and boundary lubrications. The plateau part decides about the asperity contact, while the valley part governs hydrodynamic lubrication;Negative skewness usually leads to an improvement in contact characteristics. For the same roughness height, the inclination to plastic deformation of a two-process surface is lower than that of a one-process surface. For two-process random surfaces, the plasticity index should be modified from its original formula;Two-process random surfaces are typically characterized by higher scuffing resistance, smaller wear, and coefficient of friction in mixed lubrication compared to one-process surfaces of similar roughness height. Different modes of evolution of two-process surfaces occur in dry and lubricated friction. In the initial period of life, two-process texture changes into three-process one;In further research, more modeling and experimental work should be performed to compare functional behaviors of random one- and two-process surfaces of the same roughness height. In tribological research, the Rq/Sq group of parameters should be preferred over the Rk/Sk family. In standardization, the method of determining the Rq/Sq parameters should be improved. Modeled two-process surfaces should be used more frequently in research work.

## Figures and Tables

**Figure 1 materials-15-00268-f001:**
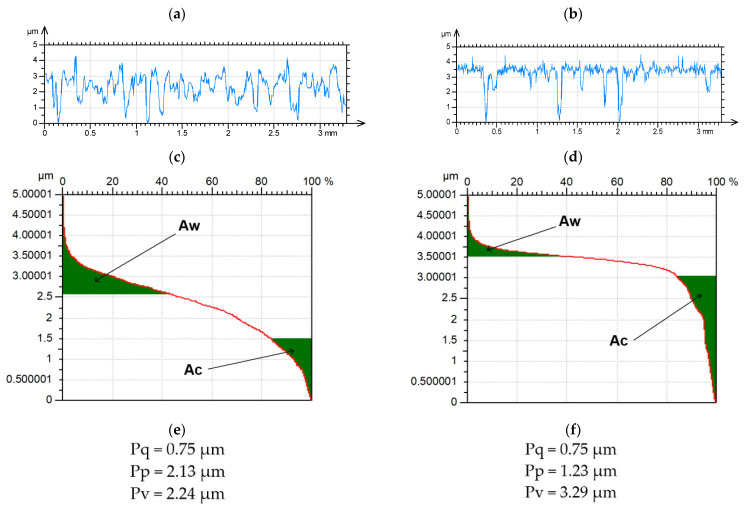
Profiles (**a**,**b**), material ratio curve (**c**,**d**) and parameters (**e**,**f**) of (**a**,**c**,**e**) one-process and (**b**,**d**,**f**) two-process surfaces.

**Figure 2 materials-15-00268-f002:**
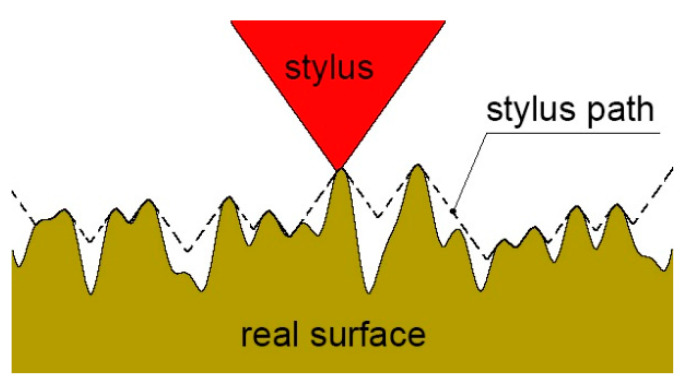
Graphical interpretation of mechanical filtration by the tip of the stylus, adapted from [24].

**Figure 3 materials-15-00268-f003:**
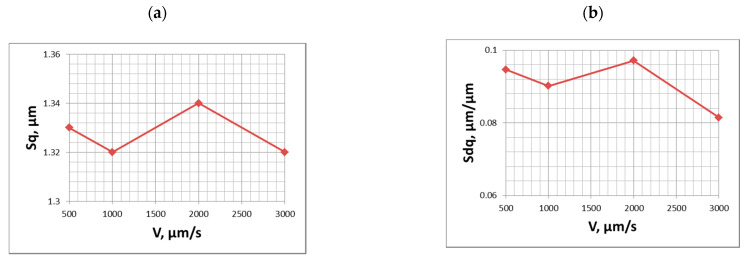
Influence of stylus speed on the values of parameters Sq (**a**) and Sdq (**b**) of the plateau-honed surface topography, adapted from [29].

**Figure 4 materials-15-00268-f004:**
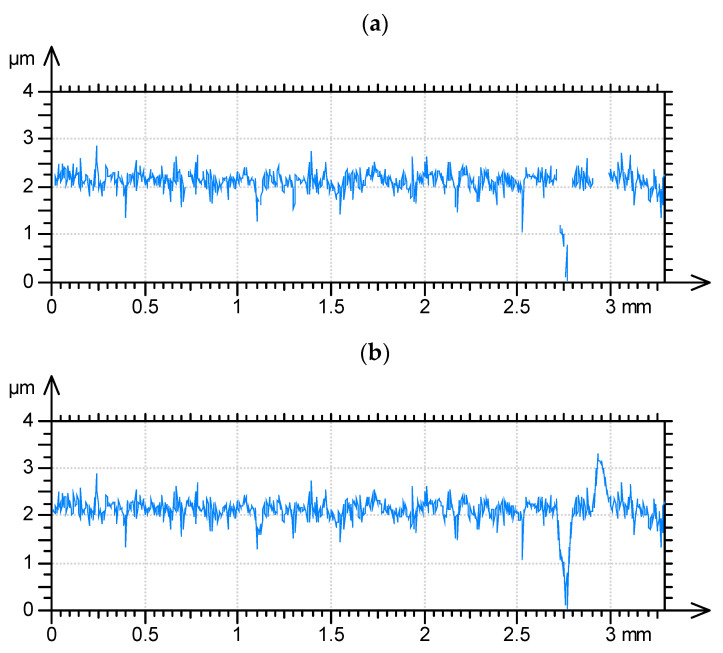
Plateau honed profile measured by white light interferometer (**a**) and with filled in non-measured points (**b**).

**Figure 5 materials-15-00268-f005:**
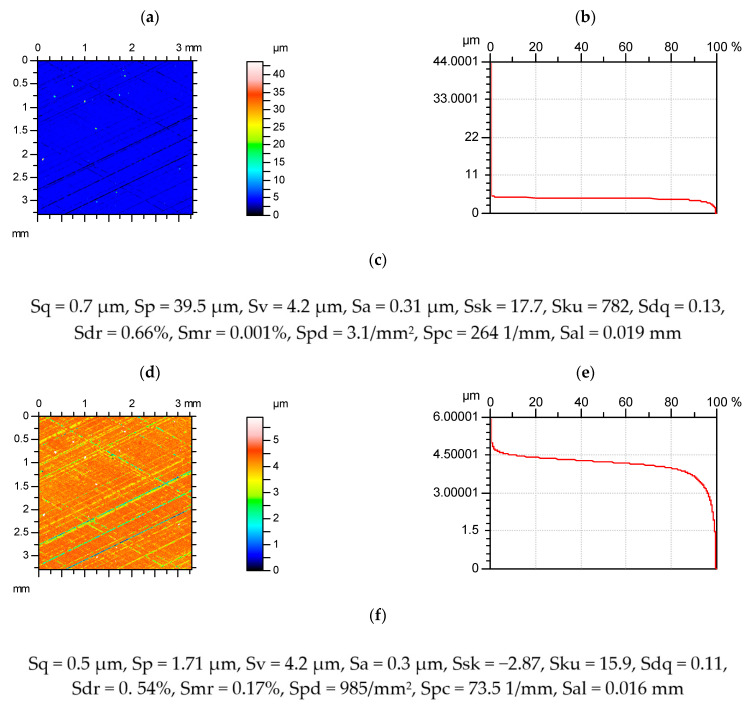
Contour plots (**a**,**d**), material ratio curves (**b**,**e**), and selected parameters (**c**,**f**) of the plateau-honed cylinder surface, with (**a**–**c**) and without spikes (**d**–**f**).

**Figure 6 materials-15-00268-f006:**
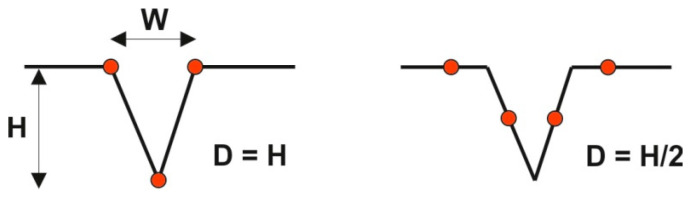
The effect of the sampling interval on the depth of the deep valley, after [45].

**Figure 7 materials-15-00268-f007:**
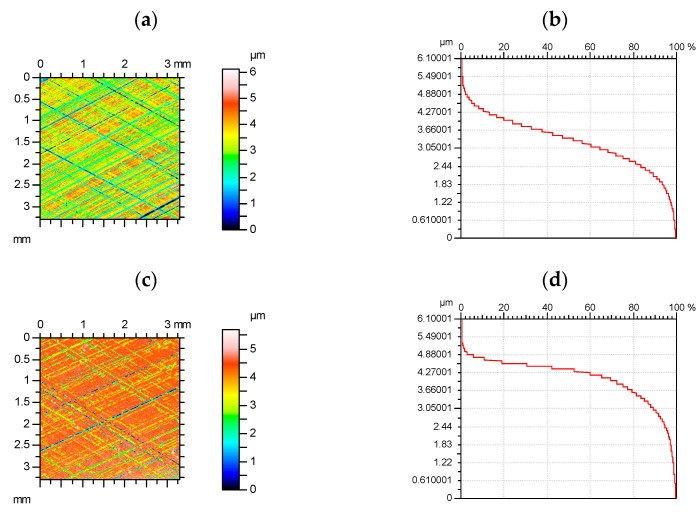
Color-code plots (**a**,**c**) and material ratio curves (**b**,**d**) of the random one-process (**a**,**b**) and two-process (**c**,**d**) surface of similar maximum height for vertical resolution of 100 nm.

**Figure 8 materials-15-00268-f008:**
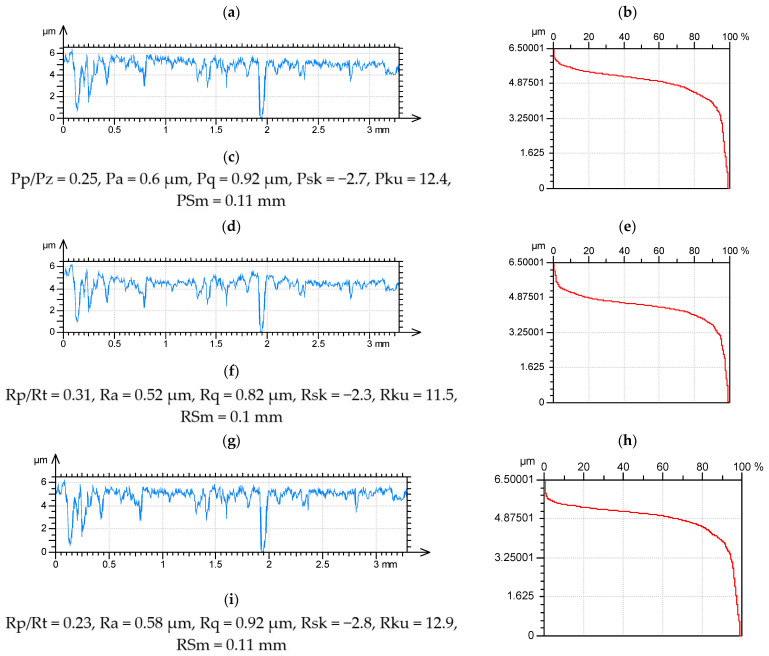
Profiles (**a**,**d**,**g**), material ratio curves (**b**,**e**,**h**) and selected parameters (**c**,**f**,**i**) of the two-process random profile (**a**–**c**), after application of the Gaussian (**d**–**f**) and robust Gaussian filter of 0.8 mm cut-off (**g**–**i**).

**Figure 9 materials-15-00268-f009:**
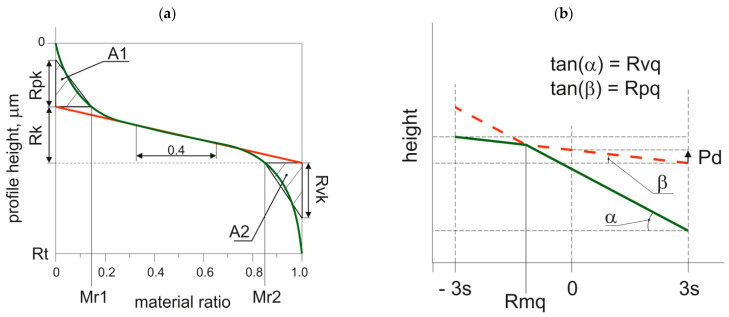
Graphical interpretation of parameters of the Rk family (ISO 13565-2 standard)—(**a**) and from the Rq group (ISO 13565-3 standard)—(**b**).

**Figure 10 materials-15-00268-f010:**
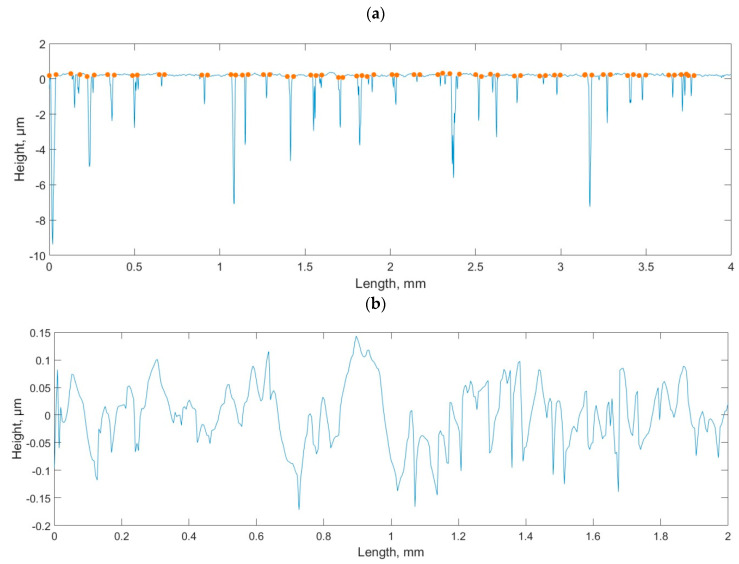
Profile with valley edges (**a**), connected plateau details (**b**), adapted from [87].

**Figure 11 materials-15-00268-f011:**
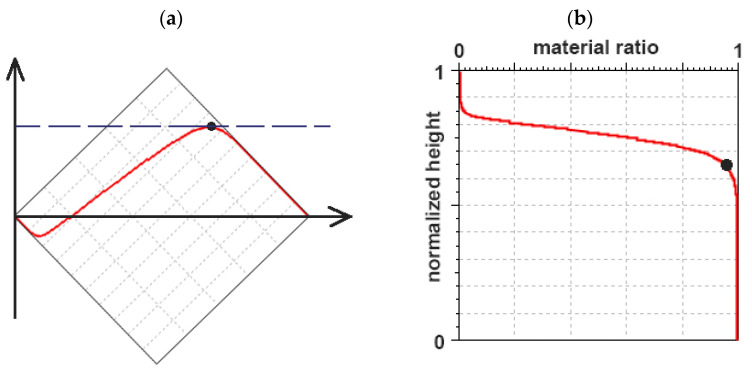
Normalized material ratio curve rotated at an angle of 45° with its highest point (**a**), this curve with a transition point (**b**), adapted from [82].

**Figure 12 materials-15-00268-f012:**
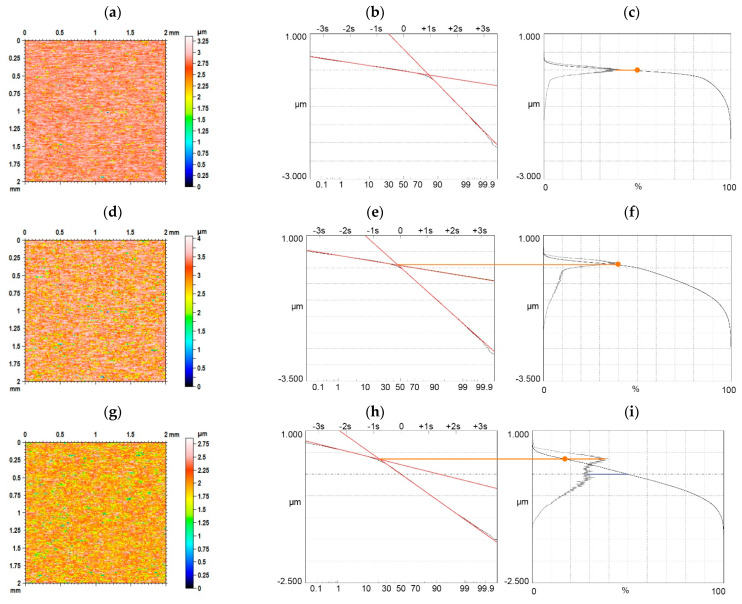
Color-code plot (**a**,**d**,**g**) probability plot (**b**,**e**,**h**), material ratio curve and probability distribution (**c**,**f**,**i**), of random two-process isotropic modeled surfaces of the Smq parameter of 80%, Spq = 0.1 µm, Svq = 0.75 µm (**a**–**c**), Smq = 40%, Spq = 0.1 µm, Svq = 0.75 µm (**d**–**f**), and Smq = 20%, Spq = 0.1 µm, Svq = 0.35 µm (**g**–**i**).

**Figure 13 materials-15-00268-f013:**
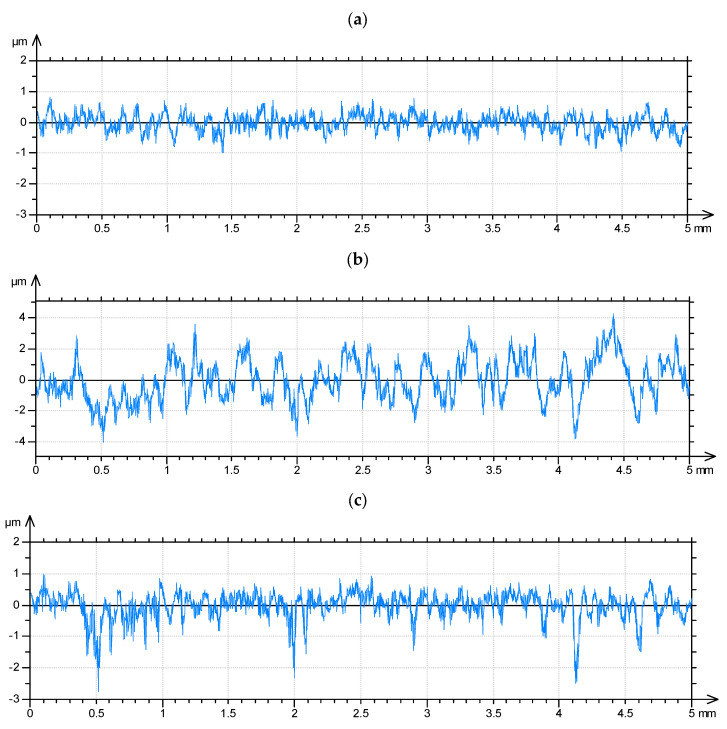
Example of generation of two-process random profile: (**a**) plateau profile: Rq = 0.3 µm, CL = 25 µm, (**b**) valley profile Rq = 1.4 µm, CL = 90 µm, (**c**) two-process profile, Rmq = 85%.

**Figure 14 materials-15-00268-f014:**
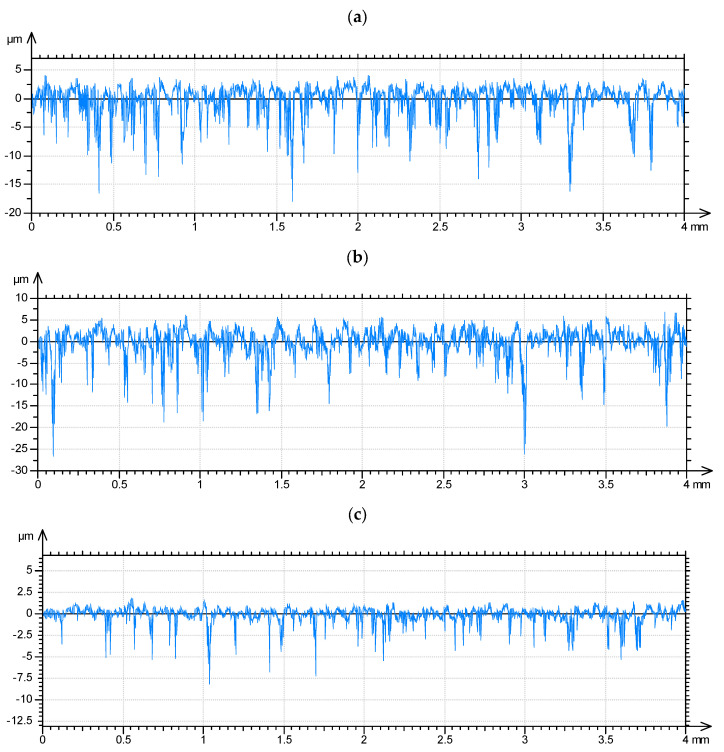
Two process random profiles: Rpq = 1 µm, Rvq = 7 µm, Rmq = 70%, Rq = 2.88 µm, Rqpr = 2.8 µm (**a**), Rpq = 2 µm, Rvq = 10 µm, Rmq = 80%, Rq = 3.69 µm, Rqpr = 3.6 µm (**b**) Rpq = 0.5 µm, Rvq = 5 µm, Rmq = 90%, Rq = 0.955 µm, Rqpr = 0.95 µm (**c**), the correlation lengths of the plateau and valley profiles were 20 µm.

**Figure 15 materials-15-00268-f015:**
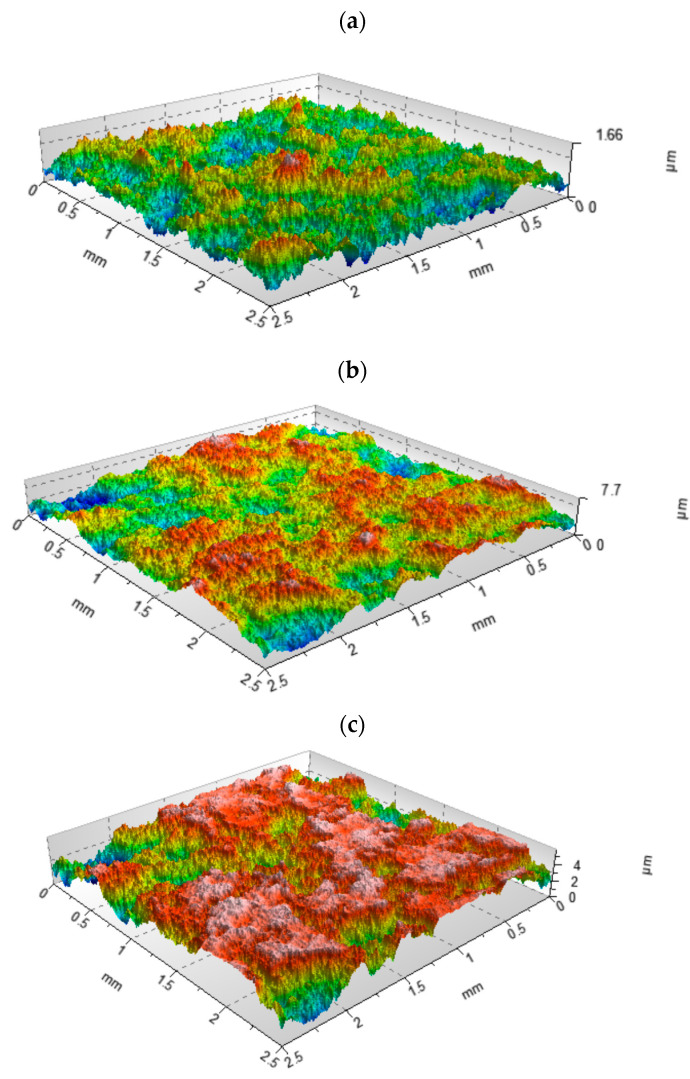
Example of generation of two-process random surface topography: the plateau surface: Sq = 0.21 µm, CL = 60 µm (**a**), the valley surface: Sq = 1.28 µm, CL = 170 µm (**b**), two-process surface: Spq = 0.21 µm, Svq = 1.28 µm, Smq = 79% (**c**).

**Figure 16 materials-15-00268-f016:**
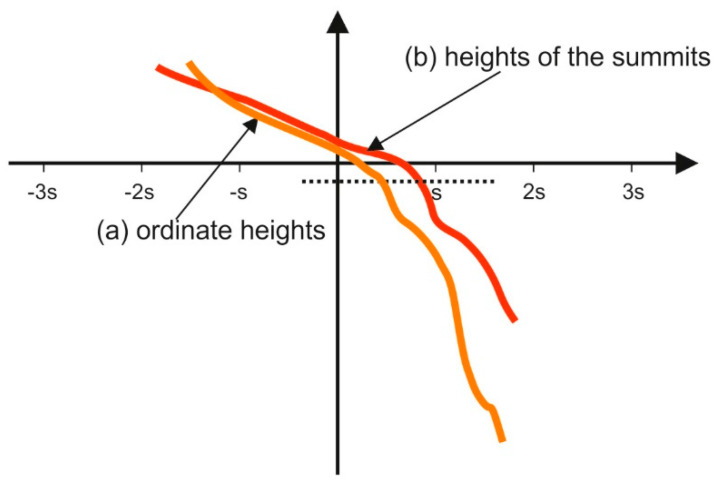
Cumulative probability distribution of the ordinate heights (a) and the heights of the summits (b), adapted from [165].

**Figure 17 materials-15-00268-f017:**
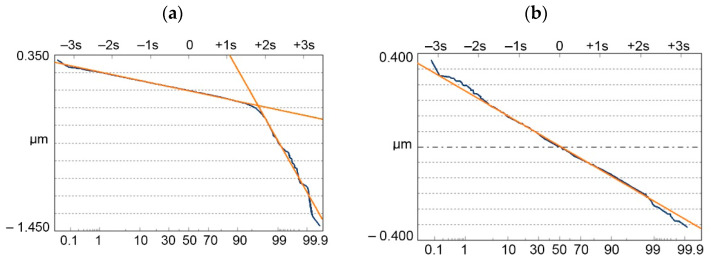
Cumulative plot of the distribution of the asperity heights of surfaces characterized by the following parameters: Spq = 0.1 µm, Svq = 1 µm, Smq = 84.13% and CLp = 25 µm and CLv = 125 µm (**a**) CLp = 12.5 µm and CLv = 12.5 µm (**b**), adapted from [167].

**Figure 18 materials-15-00268-f018:**
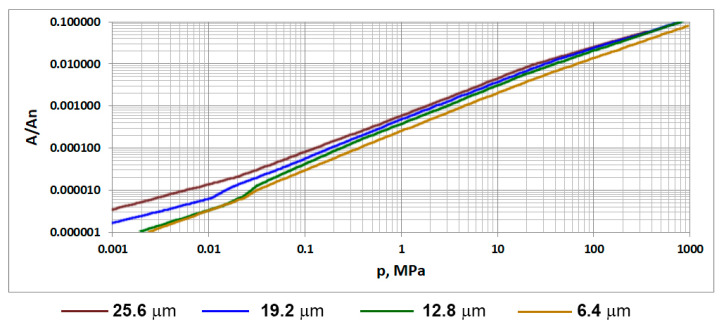
The dependence between the contact pressure and the contact area fraction for various sampling intervals, adapted from [172].

**Figure 19 materials-15-00268-f019:**
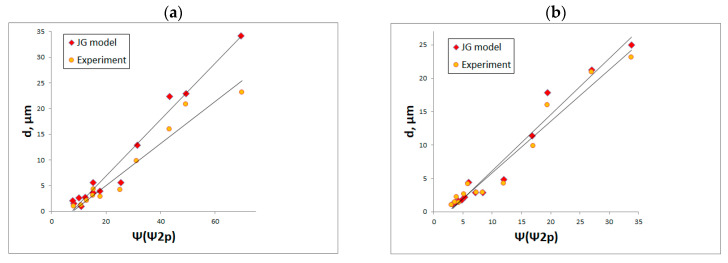
Plastic deformation and elastic-plastic deformation of one-process and two-process steel surfaces, according to the JG model [6], as a function of the plasticity index for sampling intervals of 3 (**a**) and 12 µm (**b**), for the contact load of 950 N, adapted from [174].

**Figure 20 materials-15-00268-f020:**
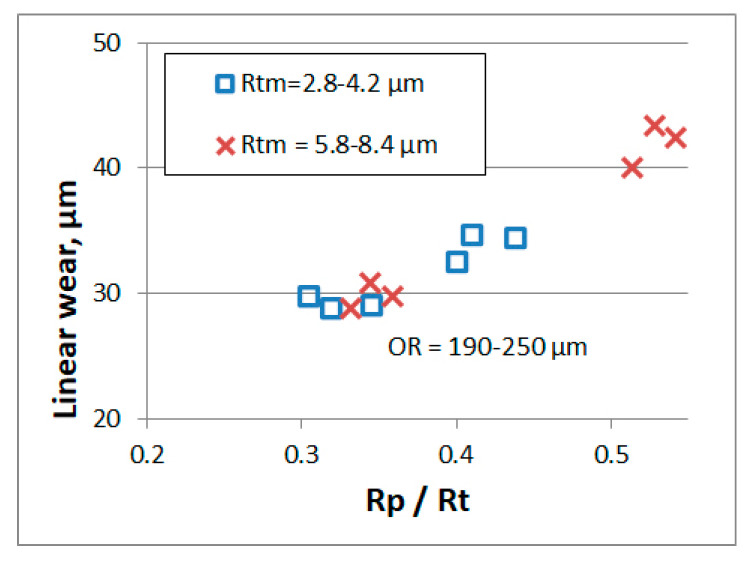
Relation between the emptiness coefficient and the cylinder wear of the engine under artificially increased dustiness conditions, for various initial values of roughness amplitude Rt parameter, adapted from [189].

**Figure 21 materials-15-00268-f021:**
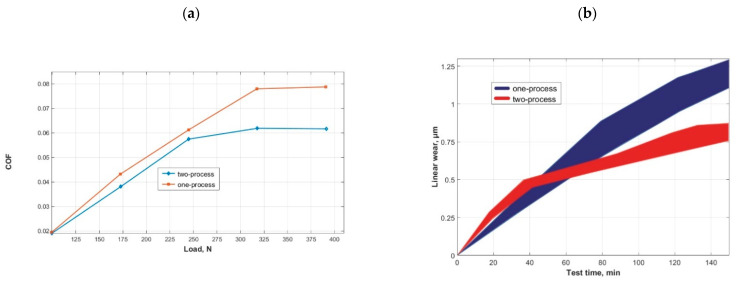
Friction coefficient as a function of load for a rotational speed of 4000 rpm (**a**), linear wear as a function of test duration (**b**), adapted from [190].

**Figure 22 materials-15-00268-f022:**
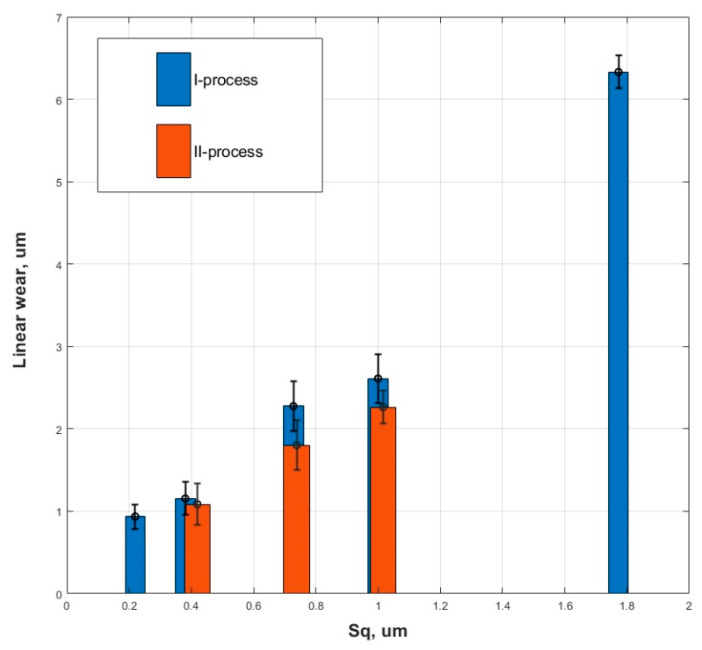
Dependence between the Sq parameter and the wear of the cylinder liner, adapted from [191].

**Figure 23 materials-15-00268-f023:**
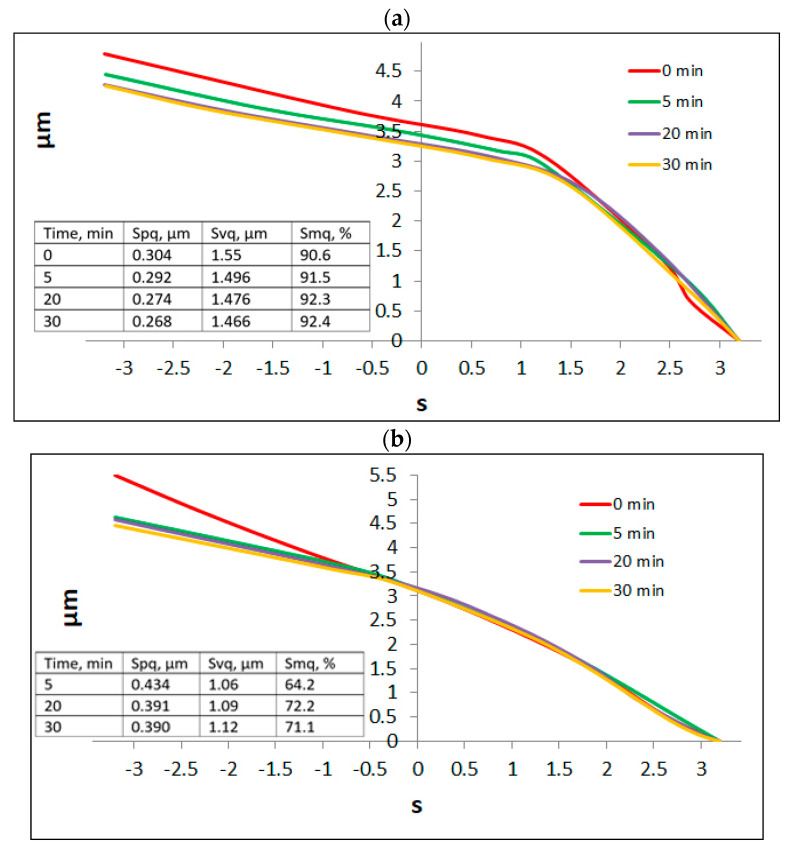
Evolutions of probability plots of the material ratio curve of the initial two—(**a**) and one-process (**b**) liners, adapted from [192].

**Figure 24 materials-15-00268-f024:**
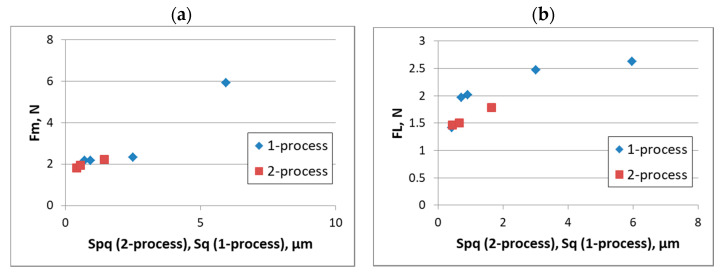
Dependence between the Sq parameter of the one-process random surface and the Spq parameter of the two-process random surface and the maximum friction force Fm (**a**) and the final friction force (**b**) adapted from [197].

**Figure 25 materials-15-00268-f025:**
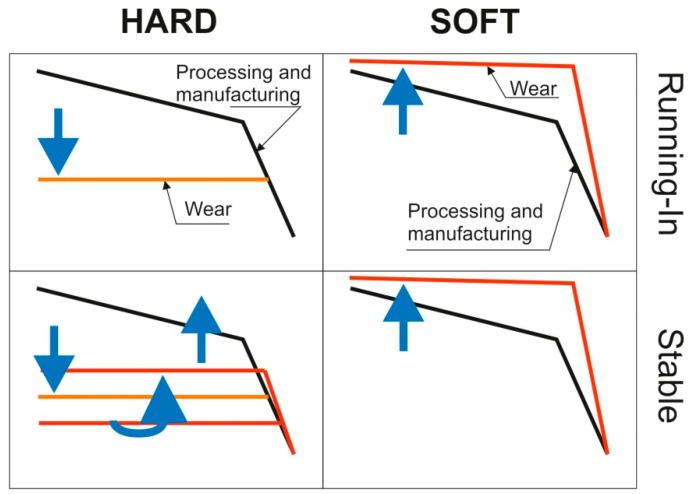
Modes of the evolution of random 2-process surface topography, adapted from [199].

**Table 1 materials-15-00268-t001:** The relations between the parameters Spq, Svq, and Smq and the parameters of summits, after [93].

	$\sigma_{s}$	$\sigma_{2ps}$	$Sd_{s}$	$Sd_{2ps}$	R	$R_{2p}$
Spq	+	+	≈	≈	−	−
Svq	+	≈	≈	≈	≈	≈
Smq	–	≈	≈	+	+	+

where + represents ‘be positively related’,− represents ‘be negatively related’, and ≈ represents ‘be nearly uncorrelated’.

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
