# Peer review of "Two-Process Random Textures: Measurement, Characterization, Modeling and Tribological Impact: A Review"

_materials, 2021, doi:10.3390/ma15010268_

Round 1
Reviewer 1 Report
Congratulations. The work is good, and it is easy to read, however, there are some concerns about your work, which can be addressed, improving the work and your understanding. Please, see the attachment.

Author Response
- In the abstract it must have a summarized sentence about the conclusions of the work.
The following sentences have been added: The nature of two-process random textures should be taken into consideration during analyses of properties of machined elements. The plateau part decides about the asperity contact, and the valley portion governs hydrodynamic lubrication.
- References 8 and 28 are not used in the text of the work. It's missing.
Improved.
- Reference 161 only appears after reference 165. You have to reorder reference 161.
Improved.
- Line 116: The authors, when they speak for the first time, in the parameters Sa, Sz, Sv and Sq, they must spell out the respective names. Or, for a better reading, after the keywords or at the end of this work, I recommend that you make a list of nomenclature, symbols, Greek symbols and abbreviations.
Nomenclature has been added.
- In chapter 2, is any standard used, namely, in the test and measurement of surface area (3D) topography?
No, the researchers typically used own procedures. However, surface texture parameters originated from specified standards (subsection 2.2).
- In chapter 2, when the authors talk about measurement errors, they do not mention whether any data processing is done after the measurements have been taken, for example: After collecting the data, the extremes are eliminated (the upper and lower peaks) and then averaged? Or is there no such care?
Data processing is done after measurements.
- The first person of the plural is used in the entire work. Please avoid this situation, because in scientific documents the first person does not be used. Please, check the words “we”. Please see line 542.
Improved.
- Should review English. Check the grammar part, namely the propositions, the plural and some verb tenses.
English has been improved.
- Format: "et al." must be in italics. Check all work.
Improved.
- The authors make 50 self-citations. It's a very high number. I leave it to the journal editor.
The present authors, especially corresponding author analysed two-process surfaces from many years. The first paper was published in 1993. Therefore this paper contains many self- citations. However, the number of citations is higher than 200. So a self-citation rate is not high. In improved paper the number of self-citation was reduced. A self-citation rate of 20% or less is characteristic of the majority of the high-quality science journals selected for coverage in Clarivate Analytics products. Now, this rate is smaller than 20%.
Reviewer 2 Report
It's a very interesting paper .It is well founded. It has many possibilities of application in practice. One suggestion: shrink figure 2
Author Response
It's a very interesting paper .It is well founded. It has many possibilities of application in practice. One suggestion: shrink figure 2.
Improved.
Reviewer 3 Report
The manuscript “Two-process random textures – measurement, characterization, modeling and tribological impact: a review” deals with an important issue related to description of two-process random surfaces.
The manuscript was submitted to the Special Issue of the journal of Materials “Advances in Surface Topography Measurement and Analysis”.
The number of reviewed papers is large and is equal to 220. Most of them are relevant and modern.
The key motivation of the authors to deal with the problem is as follows. Two-process surfaces and generally multi-process textures seem to present better functional properties than one-process ones. Plateau-honed cylinder surface is a popular example of two-process random textures. It has traces of two processes – final honing and plateau honing. During the final honing deep valleys are created, while during plateau honing the plateau smooth structure is formed.
The manuscript is well illustrated.
The manuscript consists of 4 parts:
- Introduction
- Measurement and characterization of two-process random surfaces
- Measurement errors
- Specific parameters
- Characterization of random two-process textures
- Simulation of two-process random textures
- Impact of two-process random textures
4.1 Contact mechanics
4.2. Friction and wear
- Conclusions and outlook
In fact, the manuscript is well concentrated on the topic of the review. However, the audience of the manuscript is not clear.
Regardless of the title of the special issue of the journal of Materials “Advances in Surface Topography Measurement and Analysis”, the manuscript does fall within the scope of the journal. It fits better for the journal of Surfaces (MDPI).
- The 1st and 2nd parts of the manuscript deal with surface of any kinds (mathematical and measuring issues), while the nature of real materials is not taken into account.
- The Part 4 is not directly related to Parts 2 and 3. Friction and wear problems are material oriented ones. However, this is not discussed in the manuscript.
The authors have done a great job with writing the manuscript. However, the manuscript is not relevant for the journal of Materials. It is recommended to resubmit it to a more relevant journal.
Author Response
Regardless of the title of the special issue of the journal of Materials “Advances in Surface Topography Measurement and Analysis”, the manuscript does fall within the scope of the journal. It fits better for the journal of Surfaces (MDPI).
- The 1stand 2nd parts of the manuscript deal with surface of any kinds (mathematical and measuring issues), while the nature of real materials is not taken into account.
- The Part 4 is not directly related to Parts 2 and 3. Friction and wear problems are material oriented ones. However, this is not discussed in the manuscript.
The authors have done a great job with writing the manuscript. However, the manuscript is not relevant for the journal of Materials. It is recommended to resubmit it to a more relevant journal.
Materials recently became interdisciplinary journal. In our opinion, this manuscript falls within the scope of this special issue.
“This Special Issue is an opportunity to collect information on advanced research realized by various research centres related to the measurement and analysis of surface topography in the scientific disciplines of material engineering, mechanical engineering, civil engineering, biomedical engineering, and others.
This Special Issue titled “Advances in Surface Topography Measurement and Analysis” includes high-quality original scientific articles, review articles, short messages, and case studies related to the manufacturing processes of the parts, tribological processes (including friction, lubrication, and wear), surface metrology (including modeling and simulation), and surface topography characterizing (including optimization).”
This paper is related to surface metrology, characterization and tribological processes.
Surface metrology is not materially oriented.
In revised paper information on materials used has been added to Part 4. Furthermore, information on materials for cylinder liners and on plasticity index has been added.
Round 2
Reviewer 3 Report
The manuscript has been modified, mostly in the last section. Regardless of the motivated response given by the authors, the relevance of the review paper to the journal of Materials is still a matter of discussion. However, this is a point of view of the individual reviewer. Before making the "accept" decision, the authors are welcome to motivate the paper content (idea of manuscript structuring) in the end of the Introduction section.
Author Response
The manuscript has been modified, mostly in the last section. Regardless of the motivated response given by the authors, the relevance of the review paper to the journal of Materials is still a matter of discussion. However, this is a point of view of the individual reviewer. Before making the "accept" decision, the authors are welcome to motivate the paper content (idea of manuscript structuring) in the end of the Introduction section.
Information on manuscript structuring has been added.